# Using Wikipedia to Develop 21st Century Skills: Perspectives from General Education Students

**Marvi Remmik [1,\*], Ann Siiman [2], Riina Reinsalu [2], Maigi Vija [2] and Andrus Org [3]**

[1]   Viljandi Culture Academy, University of Tartu, Posti 1, 71004 Viljandi, Estonia
[2]   Institute of Estonian and General Linguistics, University of Tartu, Jakobi 2, 51005 Tartu, Estonia
[3]   Institute of Cultural Research, University of Tartu, Ülikooli 16, 51003 Tartu, Estonia
[\*]   Correspondence: marvi.remmik@ut.ee

**Abstract:** Purpose: The aim of the study was to find out how and why students use Wikipedia and what their attitudes are towards Wikipedia as a source of information for learning. Methodology: The article is based on a quantitative study in which 381 Estonian school children participated in filling out an online survey. The questionnaire included both multiple-choice and open-ended questions. Findings: Statistical analyses and responses to open-ended questions showed that students often use Wikipedia as a primary source of information, but that their use of the site for learning tasks is guided by teachers' attitudes and perceptions towards Wikipedia. Students perceive Wikipedia as a quick and convenient source of information but are uncertain about its reliability. Under teachers' guidance, they have learnt to search for information and to be source-critical, while more attention is needed to develop academic literacy, including both text comprehension and text composition. Value: As there is still very little research on how Wikipedia is used for learning purposes in general education schools, the results of the study contribute to further discussion on the potential of Wikipedia as an innovative teaching tool for different subjects.

**Keywords:** Wikipedia in school; 21st century skills; school literacy; critical thinking; writing skills; ICT in education

## 1. Introduction

The online encyclopedia Wikipedia was created in 2001 in English. However, by now it has more than 300 language versions, 18 of which contain more than a million articles (English Wikipedia has 6.6 million articles, compared to 235,000 in Estonian). List of Wikipedias: https://meta.wikimedia.org/wiki/List_of_Wikipedias, accessed on 5 July 2023. The rapid growth of Wikipedia has been made possible by the innovative wiki technology, which greatly simplified the process of editing web pages by eliminating the need to make changes to the HTML format. This, in turn, paved the way for collaborative authoring, where millions of volunteers are involved in the compilation of articles [1]. Therefore, Wikipedia is not only seen as a multilingual online encyclopedia, but also as a community [2], where technological tools provide wider access to articles created by volunteer contributors and dedicated editors. At the same time, the Wikipedia environment does not consist only of articles, but also includes a variety of additional features such as a dictionary, a collection of citations, texts and images [3].

As one of the most popular online encyclopedias in the world, Wikipedia is also widely used by learners to acquire knowledge [4], but the use of open and freely accessible Wikipedia in teaching has created mistrust among teachers [5]. Firstly, because, although an increase in the quality and hence reliability of Wikipedia has been observed over time [2,6–8], inaccurate and biased information is found on Wikipedia [9], which students take for granted as truth due to a lack of source evaluation skills [10]. There is also a tendency to copy-paste information from Wikipedia without referencing and citing it correctly [11]. Some negative attitudes of teachers towards Wikipedia are reinforced by the

fact that a Google search returns Wikipedia as the first result and students do not bother to use other sources [10,12].

While there are schools where Wikipedia is strictly forbidden for the reasons outlined above [13], the prevailing view is that rather than rejecting Wikipedia, it should be used as effectively as possible to support learning [12,14]. The emergence of this view has been supported by changing expectations of learning and teaching: the importance of developing learners' 21st century skills has been seen as important, and this in turn has changed the tools used for learning. For example, Wikipedia-based collaborative tasks have been found to develop students' communication skills and increase the motivation to learn [14], fostering community collaboration, active citizenship and responsibility [2]. Writing, editing and contributing to articles supports information and digital literacy [6,8,15]. Wikipedia-based tasks have also been observed to have a positive impact on students' research and writing skills [16]. Although Wikipedia is more than 20 years old, it is still innovative as an up-to-date learning tool, which is why educational interventions are increasingly exploiting the pedagogical potential of Wikipedia.

Most of the previous studies on the use of Wikipedia in teaching have been conducted in the context of higher education with students and faculty (e.g., Refs. [2,4,8,14,17–19], in Estonia, Ref. [20]). Only relatively few studies have been conducted with general education students; for example, writing skills (e.g., Ref. [21]) and information literacy skills [6] have been studied in relation to Wikipedia. Likewise, there are few data on students' attitudes. An overview of Wikipedia-use habits and attitudes towards Wikipedia can be found in studies with general education students in Norway and France (see [22,23]). Indeed, it is the context of the general education school that is important to consider, as the school, led by teachers, shapes students' Wikipedia awareness and attitudes towards it. Therefore, the focus of this article is on the use of and attitudes towards Wikipedia among students in Estonian general education schools. The aim of the article is to find out how, and for what purpose, students use Wikipedia and what their attitudes are towards Wikipedia as a source of information for teaching.

## 2. Literary Review

### 2.1. Twenty-First Century Skills

Twenty-first century skills, i.e., collaborative problem-solving (including critical thinking), developing creativity, and digital and information literacy, are competences that have become particularly important in today's rapidly changing and complex societies [24]. The conscious and systematic development of these skills is also a goal of Estonian general education schools.

Solving problems and tasks, but also creating something, requires learners to work together in a way that is purposeful but also deep and meaningful, i.e., involving different cognitive processes. Designing a collaborative learning environment is a social and emotional challenge for students, who need to formulate, present and defend their ideas, discuss and exchange opposing views with peers and actively participate in group work [25]. Wikipedia is not only a new paradigm for knowledge creation and dissemination, but an online collaborative tool [12] that requires a new kind of learning skills, where participatory and collaborative activities, including building a learning community, play an important role [26].

Solving complex problems and creative tasks, including critical thinking, is considered one of the most important 21st century skills. Critical thinking is seen as an active process of reflection, evaluating facts and drawing truthful conclusions from reading and experiences [27].

Based on metacognitive mechanisms related to an individual's consciousness of the processes, actions and emotions in play [28], critical thinking increases the chances of producing a logical conclusion to an argument or solution to a problem. However, there is a lack of consensus among researchers as to what constitutes the core of critical thinking: some consider it to be a specific skill, such as the ability to correctly assess reasons, consider

relevant evidence or identify fallacious arguments, while others consider it to be a critical attitude or disposition, such as the tendency to ask probing questions. In addition, it has been debated whether critical thinking is based on a knowledge of critical thinking or on a deep and broad knowledge of a particular subject matter, which makes one a critical thinker only within that subject matter [29]. However, there is a consensus that thinking is under one's own control and that another person cannot make someone think critically. Teachers can foster critical thinking in students by providing them with clearly targeted tasks, carefully considered feedback and clear explanations, and by creating a safe classroom environment [30].

Creativity, as a key 21st century skill, needs to be developed and applied in both basic and higher education, as it motivates learning and discovery beyond the classroom [31]. Innovation, effectiveness and ethicality are key aspects of creativity [32]. To work creatively with others, you need to develop your own ideas and explain them effectively to others, be open and receptive to the opposing ideas of others and use the work and feedback you have produced in the group to develop your own ideas. Creativity and innovation are long-term processes, involving a progress of small successes and frequent mistakes [33]. Creativity also involves literacy [34]. For example, research literacy involves the ability to read, write and reason within the context of empirical knowledge [35]. Different subject teachers need to share responsibility for developing students' scientific thinking, including teaching them to use both oral and written scientific language [36].

Of the 21st century skills, digital and information literacy are receiving increasing attention in the preparation of future teachers, both in universities and in general education schools. Digital literacy, which is the ability to understand and use information from a variety of digital sources, is a vital skill and, quite literally, literacy, which involves the ability to read, write and interact with information using the technology of the age [37,38]. It means that digital literacy involves more than just handling digital information; it includes a range of technical and social skills for effective communication and content creation. On the other hand, information literacy, which involves the ability to search for information and to understand how it is created and what its value is [39], prepares learners for lifelong learning and teaches them to use information successfully [40]. While both literacies are essential, they serve different purposes: digital literacy focuses on the broader use of technology, whereas information literacy is more about the effective management and evaluation of information. Wikipedia in particular could be a tool for developing both digital and information literacy in education.

### 2.2. Wikipedia in Education: Pros and Cons

Wikipedia is seen as one way to develop 21st century skills [10,12,13]. By using Wikipedia in teaching, students can develop their collaboration skills [12,18,41], as well as their research skills, as teachers can use Wikipedia to guide research [12] and students, accordingly, can hone their research skills [14]. Creativity, too, can be developed through Wikipedia by improving (academic) literacy. Wikipedia has been used to develop first [4] and second language literacy [42], as well as digital literacy [8,10,14] and information literacy [7,18].

From the teachers' perspective, the use of Wikipedia in teaching is justified by the fact that Wikipedia is a useful teaching resource that contributes to both the modernization and quality improvement of teaching [8]. Wikipedia enables teachers to use new technologies and methodologies, and to change their attitudes towards online environments [18]. For example, students can be given the task of writing a Wikipedia article, as this can be linked to both their subject learning and their interests [10]. Writing an article is a way to work independently but also to contribute to the Wikipedia community [8]. It is an authentic writing situation, where the presence of a public addressee increases students' motivation but also their ability to manage themselves as learners [43]. In practical writing tasks, students are no longer passive recipients of information but also practical analyzers and evaluators. In addition to Wikipedia article writing tasks, existing articles can be updated,

supplemented or edited [13]. Every Wikipedia article has an associated discussion page ("Talk") where students may discuss and receive or provide information about changes to the content of an article. This kind of feedback can be educational and contributes to a more collaborative and less anonymous experience [41]. The pedagogical benefits of Wikipedia article writing depend on the subject area, the learning content and how students are engaged [44].

Students mostly use Wikipedia as an introductory and supplementary resource [45], often for quick fact-checking and background information [17,46,47]. Wikipedia is easy for students to use and understand (in most cases), guiding them to primary sources, helping them to select and narrow down a research topic [12]. In addition to ease of use, students also consider Wikipedia to be comprehensive and accurate and use it at least once a week [22,23]. According to their own perception, students have not found any errors in Wikipedia [23], although they know that the information there may be wrong and that Wikipedia is therefore an unreliable source [22]. According to students' observations, the reliability of Wikipedia depends on the purpose of its use. For example, French primary school students consider Wikipedia to be more reliable for educational purposes than for interest purposes, while high school and university students consider it to be the opposite [23].

At the same time, the use of Wikipedia for educational purposes has its own limitations due to the specificity of this online environment. On the one hand, Wikipedia's principle that anyone can add and edit content is its greatest strength, but on the other hand, it is also its greatest weakness. The fact that it is accessible to all makes Wikipedia open to vandalism and trolling. Thus, everyone has the opportunity to add inappropriate or untruthful content, while new ways to combat maliciousness are constantly being created [48,49]. In addition, some of the problems with using Wikipedia are directly related to the quantity and quality of its articles. Since users can decide for themselves which articles to add or edit on Wikipedia, the coverage of topics varies: some topics are very well covered, some are covered only partially and some are not covered at all [50]. The quality of the articles also depends on the different profiles of the users: those who have worked extensively on the sources of a topic or are experts in the field will add high-quality articles to Wikipedia, but it is difficult to prevent users who are less familiar with the topic from making additions of questionable value. There are also malicious users who want to spread misinformation, and so technologies are being developed to detect dangerous content [51].

The specific nature of Wikipedia and the varying quality of the articles on the site are often the reason for teachers' mistrust. Wikipedia as an environment may be perceived by teachers as not reliable enough, as inaccurate or out-of-date [5], as written by unreliable sources, or as not useful for academic research [12]. One of the reasons why teachers are discouraged from using Wikipedia [5] is certainly the risk of plagiarism that comes with it. Easy access to online texts and copy-paste functions contribute to students presenting the writings of others as if they were their own. Language teachers, especially English teachers, are particularly concerned about this [52]. While the issue of plagiarism has received a lot of attention in higher education contexts (e.g., Refs. [53–55]), it has received much less attention in general education (e.g., [56–58]).

Thus, if teachers or lecturers are not allowed or not willing to use Wikipedia in their teaching, neither can students. A dual attitude can be observed among students towards Wikipedia: on the one hand, they take the information it contains for granted; on the other hand, they tend to doubt its reliability [10]. As a learner, the creation of Wikipedia content is usually hampered by scarce time resources, low motivation and a lack of understanding of the task [18].

## 3. Methodology

### 3.1. Framework for the Study

The Estonian education system provides for compulsory basic education (Grades 1 to 9), followed by studies in either upper secondary school (Grades 10 to 12) or vocational

education. In order to graduate from basic school, a student must obtain a satisfactory grade in all subjects, pass the compulsory final examinations and produce a creative work based on or integrating the cross-curricular themes. However, at the end of upper secondary school, students are required to carry out research or practical work which requires, among other things, the ability to search for and critically evaluate information, to read and refer to different types of texts, to produce academic texts and to be able to cite other authors.

The development of 21st century skills is included in the eight generic competences outlined in the general part of the national curriculum for general education in Estonia and involves the development of skills that are relevant today and in the future, across all subjects and beyond the classroom [59]. In addition, the Estonian education sector aims to develop learner-centered learning and digital pedagogy by 2035, as a result of which *teachers, lecturers and trainers will be aware of the trends, opportunities, risks and uses of new technologies and will apply technology in learning in a purposeful way* [60]. Estonian teachers have a high degree of autonomy and pedagogical freedom, which means first and foremost the freedom to choose the methods they will use to teach, but also, to a certain extent, the freedom to make substantive choices [61].

The skills of Estonian students to actively participate in life after school have been assessed through the PISA tests. The main area of assessment in PISA 2018 was functional reading, which examined the acquisition of various 21st century skills (including students' ability to retrieve and evaluate information, read diagrams, find connections in text, interpret, infer). The PISA 2018 results showed that Estonian basic school students' skills were at the absolute top in Europe and in the top eight in the world [62].

As Estonian schools have all the facilities to integrate digital technologies into their teaching and students have the necessary digital skills, the question arises as to how the educational potential of Wikipedia as a collaborative online encyclopedia has been harnessed in Estonian general education schools. Therefore, the aim of this article is to find out how and for what purpose Estonian students use Wikipedia and what their attitudes are towards Wikipedia as a source of information for learning. The aim of the study is to answer three research questions:

1. What are students' Wikipedia habits?
2. What skills are students developing through Wikipedia use?
3. What are the students' perceptions of Wikipedia and what has shaped their attitudes towards Wikipedia?

*3.2. Sample*

A questionnaire-based survey of students was conducted to fulfil the objectives of the study and to answer the research questions. The schools in the sample were selected according to the principles of purposeful sampling. For this purpose, 20 schools across Estonia were contacted where, to the best of our knowledge, Wikipedia had been used in some way as part of formal education. In total, 11 schools agreed to take part in the survey. The survey was carried out with 9th and 12th grade students, as they have experience in writing longer referential texts—creative work and research papers, respectively—and it can be assumed that their exposure to Wikipedia as a source of information is therefore more varied.

The consent of all school administrations was sought to recruit the research subjects. Informed consent forms for parents were forwarded to the school liaison officer and signed and returned to the surveyors. Students completed the online questionnaire on-site at the school in December 2022 or January 2023, under the guidance of the contact person or another teacher. A total of 381 students participated in the survey (N = 124 were in Grade 9 (15–16 years old, 33%) at the time of the survey; N = 257 were in Grade 12 (18–19 years old, 67%)). In total, 43% of survey participants were male and 47% were female; 10% of respondents did not wish to identify their gender.

*3.3. Data Collection and Analysis*

The online questionnaire filled in by students contained 21 questions. The questionnaire included combining multiple-choice and open-ended questions. The questions were divided into four blocks. The questions on background information provided information on the student's gender, mother tongue and grade. Questions in the first thematic block concerned the general use of Wikipedia (including which language versions of Wikipedia are used, what information is searched for on Wikipedia and which information is preferred and what the students' experiences are with correcting, adding to and writing Wikipedia articles). The second set of questions sought data on students' own use of Wikipedia (including what they search for on Wikipedia). The third set of questions focused on students' use of Wikipedia on the initiative of the teacher (including the good and bad aspects of Wikipedia, the subjects and tasks for which Wikipedia has been used and how students have been guided in this).

The questionnaire was piloted in November 2022 with a total of twelve 9th and 12th grade students to verify the comprehensibility of the questions and the ease of understanding among the target group. Following the pilot study, the questions in the questionnaire were not changed. The questionnaire and the students' answers were in Estonian.

The data were analyzed in SPSS 20.0. Descriptive statistics, such as percentages, were used to analyze the multiple-choice questions in the questionnaire. A *t*-test (gender, grade) was used to explain differences between groups. Differences were considered statistically significant if $p < 0.05$.

Responses to open-ended questions were analyzed thematically. For this purpose, the data were entered in a table in which the responses of all students were summarized by questions. Meaningful parts of the text were then searched for in the students' answers and coded accordingly. Once the data had been coded, the codes with similar content were grouped into a corresponding category. Responses that contained different themes were first identified as themes and then grouped into appropriate categories based on content. The data were analyzed in parallel by two researchers who compared the results. When comparing the categories, it was found that there was very little variation. The agreement between the coders was 99%.

## 4. Results

*4.1. Wikipedia Usage Patterns (Frequency, Language, Purpose)*

First, we found out how often students say they use Wikipedia. The results showed that 5% of the students surveyed use Wikipedia every day, 51% at least a couple of times a week and 30% a couple of times a month. To compare the groups, we conducted a *t*-test, which concluded that statistically significant differences were present across gender and grades. For the purpose of the calculations, we treated responses as numerical (rarely/not at all = 1, a few times a year = 2, a few times a month = 3, a few times a week = 4, every day = 5). For gender, the mean is 3.73 for women and 3.46 for men ($p < 0.05$). Thus, there is a statistically significant difference in the frequency of Wikipedia use between the two groups, with female students using Wikipedia more often than male students. In a comparison of grades, the mean for the 9th grade is 3.44 and the mean for the 12th grade is 3.66 ($p < 0.05$). This implies that there is a statistically significant difference in the frequency of Wikipedia use between the two groups (albeit a weaker relationship than between males and females), i.e., 12th grade students use Wikipedia more often than 9th grade students. Statistically, this difference is significant, but the mode is "A few times a week" for both men and women. The mode for the 9th grade is "A few times a month" and for the 12th grade "A few times a week". It can be concluded from the above that female students in the 9th and 12th grade use Wikipedia more.

Next, we identified which language versions of Wikipedia students use. The different language versions of Wikipedia were reported as being used every day by 4%, at least a couple of times a week by 45% and a couple of times a month by 34% of the respondents. Of the foreign-language versions of Wikipedia, the English version is the most widely used,

with 372 respondents (98%) saying they use it. In other languages, Wikipedia was used very little by the students surveyed, for example, the Russian version was rarely used or not used at all by 315 (85%), the German version by 347 (91%) and the Finnish version by 366 (96%) of the students surveyed.

In total, 40% of the respondents admitted that they read Wikipedia in a foreign language because it is more substantial, more up-to-date and more relevant than the Estonian version. In explaining their response, several respondents also mentioned the opportunity to learn a foreign language as a positive, e.g., *I like to learn new words and I can practice the language*. The better quality of articles in foreign languages and the lack of translation errors were also mentioned.

In total, 37% of respondents said they use Wikipedia in Estonian because it is their mother tongue and it is easier to understand what they read. In the open-ended questions, it was noted that the use of Wikipedia in Estonian may be influenced by the subject matter in which it is expected to be used, e.g., *It is my mother tongue and since I tend to use it only in Estonian classes, I am expected to search for information in my mother tongue*. Also, Wikipedia in Estonian may be favored by the so-called user-friendliness, which may be expressed in the student's actions of writing down the text one by one, e.g., *Answers have to be written in Estonian and then can be written down directly*. According to 34% of the respondents, they prefer to use both the Estonian and the English Wikipedia, and the choice is made on the basis of which of the articles published in the two languages contains more of the information they need. The results showed that 85% of the respondents have searched Wikipedia for information related to various learning tasks, and 61% of the respondents have also searched for information related to their own interests. For other purposes, 3% of the respondents said they had searched Wikipedia, and the reasons given for doing so included questions arising from crosswords or a casual interest in an event or person featured in a program.

Also of interest were students' responses to the question about what activities are supported by Wikipedia. The students' responses showed that the most common activities were reading Wikipedia articles (364 respondents, 85% of all respondents), summarizing the text in their own words (304; 71%), using the text with few changes (220; 51%), researching the topic further with the help of article references (183; 43%) or copying the text (165; 39%).

Responses to the question on the learning tasks that students have used Wikipedia for are shown in Table 1.

**Table 1.** What learning tasks have students used Wikipedia for?

| | |
|---|---|
| For workbook or textbook tasks | 38% |
| Preparing various presentations | 33% |
| For research work | 25% |
| Creative work | 21% |
| Writing papers | 18% |
| Writing a discussion paper (essay) | 10% |
| Have not used Wikipedia for learning tasks | 25% |

It is important to note that as many as a quarter of the respondents to the survey indicated that they had not used Wikipedia for learning tasks, which may be an indication that quite a few teachers do not trust Wikipedia as a resource and therefore do not direct students to use it. At the same time, most of the participants (374 students) in the study indicated that they had used Wikipedia on their own without teachers' guidance, and only 7 participants in the study had not. The students' answers to the question of what activities they have performed on Wikipedia under the guidance of their teachers revealed that teachers have most often guided students to search for information and read about something on Wikipedia (269 respondents, 63% of all respondents), 147 students (34%) said

that teachers have guided them to critically evaluate and discuss what they have read on Wikipedia and 39 (9%) students had translated an article from a foreign Wikipedia into Estonian under the guidance of their teachers. While both 9th and 12th grade students said they had carried out the above, only 32 (7.5%) 12th grade students and none of the 9th grade students had written in Wikipedia and added to or corrected existing material. The reason for correcting or adding to articles was due to errors found in the articles or a lack of information in the articles. The possibility of editing a Wikipedia article also confirms a major weakness in the way Wikipedia is often not considered a reliable source: *I changed the content, I wanted to make a joke, but the moderators quickly corrected the article back to correct.* On the positive side, however, we can see here that vandalism on Wikipedia is still much more difficult than one might think. Only a few of the students who took part in the study said that they had written an article for Wikipedia, and that they had been motivated to do so by taking part in a Wikipedia article-writing competition: *I have taken part in the article-writing competition, but I have also helped edit some of the articles, because I am interested in having an Estonian version of Wikipedia, so that the information it provides is reliable and understandable to everyone in the same way.*

*4.2. Factors Supporting and Hindering the Use of Wikipedia in Teaching*

One of the aims of the survey was to find out how students perceive their teachers' attitudes towards the use of Wikipedia, and therefore what factors support and hinder the use of Wikipedia in teaching. In total, 24% of the students surveyed said that teachers had no objection to using Wikipedia, 3% said that teachers did not allow to use Wikipedia, 47% said that some teachers did and some did not and 10% said that they did not know. Teachers do not explicitly forbid students from using Wikipedia for learning tasks, but they do recommend that students use more trustworthy sources: *They strongly advise against using Wikipedia, but since we are already in 9th grade and we have to be responsible for where we get our information from, teachers cannot explicitly forbid us to use Wikipedia. Generally, they still don't like it when someone uses it as a source.* In some cases, the teacher does not recommend using what is written in the Wikipedia article, but they do recommend further research into the sources following the article. According to some of the students surveyed, the teacher is not interested in where the information comes from and focuses more on their correct answers.

The students' answers to the question on which subjects teachers have guided them to use Wikipedia show that this has been performed by teachers of all subjects, with the highest number of teachers of history (143 respondents; 38% of all respondents), geography (131; 34%), music (113; 30%) and literature (102; 27%). Teachers of physical education (35 respondents; 9% of all respondents), arts and crafts (34; 9%), mathematics (33; 9%) and technology (20; 5%) are the least likely to have used Wikipedia.

The three most important themes or positive aspects of Wikipedia were mentioned as good points: (1) easy and quick access to information: *teachers consider Wikipedia to be a quick and convenient source of information; Wikipedia is often the first answer on Google and contains the most relevant information at the top of the page;* (2) clarity of content: *easy access to information; not too comprehensive/extensive; language is appropriate for students;* and (3) constant updating of information and content: *constantly changing; articles are updated as new information becomes available.* In terms of the accessibility of information, the ability to access primary sources and other sources on the same topic is also mentioned as a positive: *good way to find other sources on the same topic.* However, there are even more students in the survey who say that *teachers don't really talk about the good things about Wikipedia very often.* Moreover, *I think teachers don't see any good things about using Wikipedia.* The results of the survey showed that there were conflicting opinions about the reliability of the information on Wikipedia. For example, some of the respondents to the survey said that their teachers found the information on Wikipedia to be accurate and reliable: *easy to access and read; mostly accurate and checked; often referenced to primary sources.* In several cases, the reliability of information on Wikipedia was attributed to the specificities of the subject

area: *In science, the knowledge of facts is solid and in these subjects Wikipedia provides a good and comprehensive picture.*

The students' opinions on what their teachers consider to be the drawbacks of Wikipedia were also mainly related to the reliability of the information on Wikipedia. Students' responses were dominated by opinion: *My teachers have argued that no researcher uses Wikipedia for referencing in their research because it is not a reliable enough source for accurate information.* Wikipedia's credibility was also undermined by the fact that the information it contains can be added by all users, which in turn can mean that the information is potentially inaccurate: *There are facts that have not been checked; Sometimes the text of the article omits references to primary sources. Anyone can change the content and thus Wikipedia's science-based credibility is diminished.* The unreliability of information on Wikipedia was also associated with the existence of outdated information. However, on the organizational side of Wikipedia, the template at the beginning of the article, which is a warning to the reader about the potential unreliability of information, was highlighted as a positive: *Those with no or few references and no discussion. Occasionally, there is also a large exclamation mark at the beginning of the article indicating problems with the content.* One student also pointed out problems with the content of articles in terms of possible bias depending on the views or preferences of the writer of the article: *It is biased and historical events are a good example.* Unfortunately, it is not possible to tell from the student's response whether this was the student's own opinion or that of his/her history teacher.

Problems with Wikipedia that were brought to the students' attention by their teachers included factual and spelling errors in the articles and the fact that *not all articles are updated and moderated often enough.* The uneven quality of Wikipedia articles in Estonian was also mentioned: there are some topics where the articles are substantial and in-depth, and others where they rely on a single source. On the pedagogical side, a problem identified by teachers was that they often found that students did not go deeper into the content of the article but copied the text for their own work.

*4.3. Challenges for Students Using Wikipedia*

In their experience of the challenges of using Wikipedia, students repeatedly mentioned that information on Wikipedia can be outdated or is not always correct. For example, students have noted discrepancies between the dates given in the textbook (e.g., historical events) and those given in the relevant Wikipedia article. This, in turn, is linked to students' perceptions of the dubious reliability of the information contained in Wikipedia: *Often the Wikipedia text is not up to date and inaccurate information can be obtained. Also, Wikipedia texts are accessible to anyone and can be written about by anyone and about anything.* In addition to factual errors, students have noticed misspellings and other inaccuracies in Wikipedia articles.

Several students have also highlighted as a problem the fact that, because Wikipedia articles are long and sometimes complex in content, finding the information they need is quite a challenge for them. *Since most of the Wikipedia texts are long, I haven't found the right information.* However, the students have also compared the content of the Estonian and English articles and found that while the Estonian article does not contain the information they need, the English article usually does. Here, however, there may be problems with rephrasing the text and conveying the meaning, as there is a risk that some of the necessary information will be lost in translation.

As Wikipedia is often the first source of information displayed when searching for information, students will also read the information it contains first and use it for various learning tasks. However, this is a problem according to the students, as many of the students' work is then very similar: *All the students use the same source, which becomes obvious when the work is compared. The wording is very similar and the information is written in the same order.*

The authorship and citation of Wikipedia articles has also been cited as a problem by students, as the authors' names and the date of the article are not visible. This, according to

the students, makes it difficult to reference Wikipedia. At the same time, Wikipedia clearly states that it is not a primary source of information. What can be cited are the references at the end of each Wikipedia article. However, another important concern related to authorship can be identified from the students' views on the problems with Wikipedia. It appears that students copy directly from Wikipedia into their own work and are more concerned about solving technical problems than about the risk of plagiarism in their own work: *if you copy directly from Wikipedia, the links come with their words*; *it's cumbersome to take the links away*. Several students admit that they find it difficult to reference and paraphrase the text, which makes them feel that the wording of the article is the best they can do: *The biggest problem I have encountered so far in using Wikipedia is probably related to referencing the text. Often I feel that the best wording has already been used in Wikipedia, and then I have to think long and hard about how best to use the knowledge from the article in my own words.*

*4.4. Teachers Using Wikipedia to Support Learning*

We asked students to describe how teachers have supported them in using Wikipedia. In both the 9th and 12th grade, students mentioned source criticism and the ability to assess the credibility of the sources on which an article is based the most. *The teacher has also advised us to check the sources cited and evaluate the page, explaining how to spot misinformation.* They have also been advised by their teachers to compare the Wikipedia article with other sources.

A number of 12th grade students referred in their answers to electives or research support courses at school, which have also provided them with different insights into what to look out for when reading articles: *There is an elective course on "Information Reading and Source Criticism" at school, where the teacher talked about how to analyse a source and its credibility. Nobody has given me much guidance really in reading and searching for information.* Participants in the study also included those who had analyzed the history of Wikipedia pages with a teacher and learnt to draw conclusions from it: *We have looked at different Wikipedia pages together with the teacher and examined them critically—what is missing, what indicates that the article is not reliable, how many sources there are, how many references there are, when the article was created, who wrote it*, etc.

A few students mentioned that they have been instructed and guided by their teachers to change the content of articles when they have found errors or gaps in content: *Teachers have encouraged us to change information if it is really incorrect and warned that the information is not correct. We have also discussed misinformation in class.* The teachers have also instructed them on how to translate Wikipedia articles: *When translating, the teacher has instructed us to use a dictionary and explained how to cite correctly /.../ the teacher has forbidden copying; how to correctly formulate the information we have read in our own words.*

According to a number of students, they are not advised by their teachers to use Wikipedia as a source and are therefore not instructed on how to work with Wikipedia: *I have been instructed by teachers to use books. I have not been instructed by any teacher to use Wikipedia for any learning task.* However, there were also students who said that teachers have become more accommodating towards the use of Wikipedia: *Teachers have never really guided us to use Wikipedia. They advised against it for a long time, so there was no guidance. Now that it has become more popular and teachers are more supportive, it is expected that we can already use Wikipedia properly ourselves.*

## 5. Discussion

The development of 21st century skills such as problem-solving, critical thinking, creativity and digital and information literacy must be consciously and systematically promoted in general education schools [24]. Studies have shown that Wikipedia, the most widely and frequently consulted online encyclopedia by learners, can be successfully used to develop the above-mentioned skills (see e.g., Refs. [2,4,12,14,19,41]). As most of the research on Wikipedia in education has been conducted in the context of higher education, we carried out a survey in Estonian general education schools. The aim of the survey was

to find out how and for what purpose students use Wikipedia and what their attitudes are towards Wikipedia as a source of information for learning. In the following, we discuss the main findings.

First, we set out to uncover the Wikipedia habits of students. While previous studies have shown that students in European general education schools use Wikipedia at least once a week [22,23], our results also showed that 56% of the students surveyed use Wikipedia on a weekly basis for different learning tasks and for personal use. When completing learning tasks, they use Wikipedia to find quick answers for assignments such as workbooks and textbooks, i.e., they mainly read Wikipedia, and less frequently correct, add to or translate articles.

The results of the study showed that 12th grade students use Wikipedia more often than 9th grade students, and that the frequency of use was also higher among female students. The higher use of Wikipedia in the 12th grade can be explained by the fact that in upper secondary school, students are expected to be more independent in their learning tasks, which means that they have to search for different information more often during their studies. However, regarding why girls' use of Wikipedia is higher compared to boys', we did not find an answer in the present study and this question therefore needs further attention.

What was interesting from the results of the survey was that students use both Estonian and English Wikipedia with equal frequency. There was no statistically significant difference between the Wikipedia usage habits of 9th and 12th grade students. The extensive use of the Estonian Wikipedia can be considered understandable due to it being students' mother tongue, but the frequent use of the English Wikipedia indicates that the students themselves consider their language skills to be quite good, and therefore do not shy away from searching for information from a foreign source when necessary. While in a study conducted in Norway, school students used Wikipedia in other languages (including Spanish, Swedish, Danish, French, German) in addition to Norwegian and English [22], in this study only a few students mentioned the use of Wikipedia in other languages. This may be due to the fact that in most Estonian schools the first foreign language taught is English (and to a much lesser extent, German), which means that students have a better command of English than other foreign languages.

The second research question focused on what skills students develop through Wikipedia use. The results of the survey showed that most participants used Wikipedia on their own initiative to search for information for different assignments or personal interest. Teachers most often directed students to search for information and read about something on Wikipedia. This shows that of all 21st century skills, Wikipedia use is most likely to support the development of digital and information literacy skills for both students and teachers. However, as the focus is mainly on information retrieval, but tasks requiring writing, editing or correcting Wikipedia articles are rarely given, it can be suggested that the full potential of the Wikipedia environment for developing digital and information literacy skills is not being exploited.

Students can develop and learn academic writing skills (e.g., referencing) through Wikipedia use, and therefore teachers of all subjects should provide students with tasks within their subject that support the ability to comprehend text and express it in their own words. Consideration should also be given to the age at which the skills are to be developed in learners. For example, at the basic school level, students could be allowed to refer to Wikipedia, with a focus on teaching referencing and citation skills, and only at the upper secondary level could students be required to go to primary sources.

Ethical issues were raised in relation to the development of students' academic literacy, bringing to the fore both authorship and plagiarism. Studies in the context of higher education have shown that plagiarism is often not intentional, but a problem of lack of awareness among learners of both plagiarism and ways to avoid it [52]. In the case of Wikipedia, which is a reference work in its nature, i.e., material compiled from different sources, each sentence of an article should indicate from which source it is referenced. Thus,

if the students learn something from a Wikipedia article that they would like to use in their work, they should go back to the original source and read and refer to it. However, the students' answers show that most of the time they do not go back to the original source but prefer to refer to Wikipedia. There is also a concern that Wikipedia does not have an author to write in the reference. Students also admit that it is difficult for them to convey in their own words what they find in Wikipedia, and even more so that they feel that the wording of the Wikipedia article is the best possible and are afraid to rephrase what is in the article. Students may also find it difficult to translate an article into a foreign language and have a consequent fear of paraphrasing what the article says. They often opt for the copy-paste technique, which students say could make their work closely resemble the original and heighten the risk of plagiarism. Similar problems have been cited in a number of previous studies; for example, a study in New Zealand found that nearly a quarter of student submissions contained plagiarism, even though the nature of plagiarism and ways to avoid it had been explained to the subjects beforehand [63].

The results of the survey suggest that Wikipedia's shortcomings can also be an important source of learning for students, with support from teachers. For example, collaborative class discussions on analyzing the information in a Wikipedia article, as well as discussions on when the information in the article has been updated, support the development of students' 21st century skills such as digital literacy, information literacy, critical thinking and analytical skills. The results of the study suggest that in this respect, 12th grade students have benefited from important support in the form of school electives on information literacy, as well as courses that support compulsory research writing. Unfortunately, such courses are not offered to students in basic schools, which means that students may miss out on important information and support from teachers in basic school.

The third research question asked what students' perceptions of Wikipedia are and what has shaped their attitudes towards Wikipedia. Earlier studies have shown that students' use of Wikipedia can be influenced by teachers' and other students' attitudes toward Wikipedia. This is how Garrison's [17] study showed that first-year students' ratings of Wikipedia were influenced by their high school teachers, college faculty and classmates. Positive influences corresponded with positive ratings of Wikipedia, and the reverse. Thus, we can see from the results of the study that the teacher plays an important and main role in shaping students' attitudes towards Wikipedia. According to the results of our study and the results of Garrison [17], it can be said that the age of the learner has little influence on the use of Wikipedia.

The results of our study indicated that teachers are still skeptical about the value of Wikipedia as a resource and their attitudes influence students' choice of learning materials—if teachers do not consider Wikipedia to be a reliable source, students, too, are more wary of using it. The importance of the teacher as a designer and facilitator of the learning process is confirmed by the results of previous studies (e.g., Refs. [22,23]), as well as the present study. For example, as many as a quarter of the students in our study did not use Wikipedia in their learning tasks. More importantly, there are still teachers in schools who do not recommend the use of Wikipedia as a source, as articles can be created or corrected by anyone who wishes to do so, and both factual and spelling errors have been found in articles. In a number of cases, some respondents admitted to having made a joke themselves by changing the content of some articles and adding misinformation, and then checking to see if and when the errors were corrected.

The results of the survey showed that teachers of various subjects use Wikipedia very differently in their teaching, with the most common use being in history and geography lessons, according to students. The specific nature of these subjects means that the results of the study are understandable, but the results of our study did not allow us to understand why, for example, technology and mathematics teachers use Wikipedia so rarely in their teaching. In order for Wikipedia to be more widely used in teaching and for teachers to understand the innovative and instructive possibilities that Wikipedia offers both for themselves and for their students, it would be necessary to provide teachers with relevant

training and guidance materials that would save teachers' time and also introduce the possibilities that Wikipedia offers for the development of students' different subject and generic competences. A good starting point for this is provided by the Wikimedia Foundation's program "Reading Wikipedia in the Classroom," which includes a three-module Teacher's Guide for engaging with Wikipedia in a secondary context, as well as teaching materials created by Wikimedia Eesti (https://wikimedia.ee/oppematerjalid/, accessed on 4 January 2024). Teachers themselves should also be encouraged to improve the content of articles, add new content or create new content where necessary, as they can help to make Wikipedia more reliable in the future.

The current results cannot be generalized to all students in Estonia, as the survey included schools and students who were previously known to have used Wikipedia to some extent in their teaching. Nonetheless, the results provide an insight into the Wikipedia-use habits of Estonian school students and the factors that have supported or hindered students' use of Wikipedia in their studies. The questionnaire developed for the current study, however, provides an opportunity for further research to expand the sample population by including students who are not predisposed to Wikipedia engagement and to gain a more representative overview of how Wikipedia is used for educational purposes in Estonian general education schools.

**Author Contributions:** Conceptualization, M.R., A.S., R.R., M.V. and A.O.; methodology, M.R., M.V., R.R., A.S. and A.O.; formal analysis, M.R.; investigation, M.V, R.R. and A.O.; data curation, M.R., M.V., R.R., A.S. and A.O.; writing—original draft preparation, M.R., A.S., R.R., A.O. and M.V.; writing—review and editing, M.R., A.S., R.R., A.O. and M.V.; project administration, R.R.; funding acquisition, R.R. All authors have read and agreed to the published version of the manuscript.

**Funding:** This research was funded by Wikimedia Foundation, Inc., grant number G-RS-2204-08618.

**Institutional Review Board Statement:** This study was conducted in accordance with the Declaration of Helsinki and approved by the Ethics Committee of University of Tartu (protocol code 368/T-10, 19 September 2022).

**Informed Consent Statement:** Informed consent was obtained from all subjects involved in the study.

**Data Availability Statement:** Data are unavailable, due to ethical restrictions.

**Acknowledgments:** The authors thank the study participants for voluntarily participating in the study and providing such useful insights.

**Conflicts of Interest:** The authors have no conflicts of interests. The funders had no role in the design of the study; in the collection, analyses, or interpretation of data; in the writing of the manuscript; or in the decision to publish the results.

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
