# Peer review of "Using Wikipedia to Develop 21st Century Skills: Perspectives from General Education Students"

_education, doi:10.3390/educsci14010101_

Round 1
Reviewer 1 Report
Comments and Suggestions for Authors
Dear authors:
Thank you for the opportunity to review this compelling research study, which contributes to discussions of secondary students' attitudes and uses of Wikipedia in the context of Estonian general-education schools. Overall, I found the article manuscript to be well-written, and the study conducted to be methodologically robust, sound, and productive. Indeed, there are a number of insights reported here that are novel to our understanding of this group's engagement with and dispositions toward Wikipedia as a supplemental platform for learning. I was especially intrigued by the finding that self-reported female students spend more time on Wikipedia than their male peers.
Additional important findings include the specific ways teachers are instructing students to use Wikipedia (e.g. to assess source credibility), and the finding that upper-level secondary students are more likely to use Wikipedia than their 9th grade peers.
Some revisions requests for further improving a solid article manuscript:
1) Do more to expand on recommendations for productive engagement with Wikipedia in the secondary context. At one point in the manuscript, the authors suggest that "at the basic school level, students could be allowed to refer to Wikipedia, with a focus on teaching referencing and citation skills, and only at the upper secondary level could students be required to go to primary sources." I appreciated this kind of practice recommendation and would like to see more of this if possible as outcomes of the research findings. I also think that Estonian public school teacher's pedagogical freedom/autonomy, as described in this article manuscript, could be a significant factor in more wide-scale training of teachers and adoption of Wikipedia-based educational practices in the secondary context. But I would like to hear more from the authors about how such a program could be carried out on a wider scale. Would the authors be willing to provide a brief paragraph on how these research findings might aid in the planning of a teacher-training program similar to the "Reading Wikipedia in the Classroom" program created by Wikimedia Education?
2) The above recommendations for teaching might be paired with additional recommendations for future research. One significant limitation of this study, which the authors do acknowledge, is that the sample population studies has already been predisposed to Wikipedia engagement. A clearer indication of how future research might cast a wider net among the secondary/general-education in Estonia would be appreciated in a concluding paragraph or section of the paper.
3) Because a good portion of the sample population studies were 12th graders, I do think there is relevance here for how we might consider postsecondary / undergraduate habits and attitudes, at least as they transition into the college or university context. Are there studies on Estonian undergraduate engagements with Wikipedia? Perhaps not, but I would recommend that a small section be added that compares your findings to studies that have been conducted on first year undergraduate students' usage/attitudes concerning Wikipedia in other cultural/geographical contexts. This type of comparison might help us, at the very least, understand how advanced secondary students differ (or do not differ) from early postsecondary students in habits and attitudes. Some possibilities for sourcing this discussion include:
Garrison, J.C. Getting a “Quick Fix”: First-Year College Students’ Use of Wikipedia. FM 2015, doi:10.5210/fm.v20i10.5401.
Selwyn, N.; Gorard, S. Students’ Use of Wikipedia as an Academic Resource — Patterns of Use and Perceptions of Usefulness. The Internet and Higher Education 2016, 28, 28–34, doi:10.1016/j.iheduc.2015.08.004.
These revisions would go a long way in improving what is already a strong manuscript, and I look forward to seeing this in print!
Author Response
We would like to thank the reviewers for the valuable and constructive comments. In accordance with the suggestions, we have done revisions in the article - all changes and additions are easy to find as they are made in red in the manuscript.
Best wishes, Marvi Remmik

Reviewer 2 Report
Comments and Suggestions for Authors
1. Critical thinking is a metacognitive process that increases the chances of producing a logical conclusion to an argument or solution to a problem.The part where Critical Thinking is discussed need to include some references on Metacognition for exaple see these:
Zhou, M., & Lam, K. K. L. (2019). Metacognitive scaffolding for online information search in K-12 and higher education settings: a systematic review. Educational technology research and development, 67(6), 1353-1384.
Yamamoto, Y., Yamamoto, T., Ohshima, H., & Kawakami, H. (2018, May). Web access literacy scale to evaluate how critically users can browse and search for web information. In Proceedings of the 10th ACM Conference on Web Science (pp. 97-106).
2) The distinction between Digital Iteracy and Information Literacy is not clearly presented.
Clarify that digital literacy involves more than just handling digital information; it includes a range of technical and social skills for effective communication and content creation Emphasize that while both literacies are essential, they serve different purposes. Digital literacy focuses on the broader use of technology, whereas information literacy is more about the effective management and evaluation of information.Emphasize that while both literacies are essential, they serve different purposes. Digital literacy focuses on the broader use of technology, whereas information literacy is more about the effective management and evaluation of information.
3) "A total of 381 students participated in the survey (N=124 were in grade 9 (33%) at the time of the survey; N=257 were in grade 12 (67%))."
Better specify the age range of students, not all international journal readers knows what age are 9th and 12th grade students.
4) It is not clear how many students said they use Wikipedia on their own and how many within an educational activity led by the teachers. This can lead to different results interpretation.
5) "The three most important themes or positive aspects of Wikipedia were mentioned as good points..."
Please enumerate 1), 2) and 3) points because it is not clear in the following text.
6) "The authorship and citation of Wikipedia articles has also been cited as a problem by students, as the authors' names and the date of the article are not visible. This, according to the students, makes it difficult to reference Wikipedia"
Wikipedia clearly states that its *not* a primary source of information. What can be cited are the references at the end of each Wikipedia article!
7) it would be useful to at least mention the fact that each Wikipedia article has an associated discussion page and participation in the discussion is a very educational activity because it allows you to discuss/delete the changes made.
8) "Only a few of the students who took part in the study said that they had written an article forWikipedia, and that they had been motivated to do so by taking part in a Wikipedia article writing competition:"
"Teachers themselves should also be encouraged to improve the content of articles, add new content or create new content where necessary, as they can help to make Wikipedia more reliable in the future"
This would have been a very important point to discuss precisely because of the educational value of writing an article not only by teachers but students *together* with teachers to guarantee quality content and to develop communications and Critical Thinking skills. See for example:
Petrucco, C. (2018). Wikipedia in university courses: teaching practices and educational benefits. Research on Education and Media, 10(2), 10-16.
9) in the part of the discussion of the results, it would be useful to list the possible skills acquired in a more structured and clear way.
10) However, the questionnaire lacks specific questions or teachers' observations on the skills that students may have acquired while using Wikipedia (writing skills, critical analysis...etc.) So, how can you say that they have acquired them?
Author Response
We would like to thank the reviewers for the valuable and constructive comments and feedback. In accordance with the suggestions, we have done changes and additions in the article. All changes are easy to find as they are made in red in the manuscript. Best wishes, Marvi Remmik

Round 2
Reviewer 2 Report
Comments and Suggestions for Authors
The conclusions should be presented in a more structured way and in points, in order to adequately answer the research questions.
Author Response
Dear reviewer,
Thank you for your valuable comments. We have considered all the reviewer`s suggestions and restructured the discussion chapter where we have taken the research questions as the basis. We have also organized the references used in the article. All changes are colored green.
Best wishes
Marvi Remmik
